# PCB-Based Planar Inductive Loops for Partial Discharges Detection in Power Cables

**DOI:** 10.3390/s23010290

**Published:** 2022-12-27

**Authors:** Sinda Kaziz, Pietro Romano, Antonino Imburgia, Guido Ala, Halim Sghaier, Denis Flandre, Fares Tounsi

**Affiliations:** 1SI2E Laboratory, École Nationale d’Ingénieurs de Sfax, Université de Sfax, Sfax 3038, Tunisia; 2Faculté des Sciences de Monastir, Université de Monastir, Monastir 5019, Tunisia; 3L.E.PR.E. H.V. Laboratory, Department of Engineering, University of Palermo, 90128 Palermo, Italy; 4SMALL Group, ICTEAM Institute, Université catholique de Louvain, 1348 Louvain-la-Neuve, Belgium

**Keywords:** inductive loops, PCB-based inductors, partial discharges, defects detection, HV power cables

## Abstract

Partial discharge (PD) diagnosis tests, including detecting, locating, and identifying, are used to trace defects or faults and assess the degree of aging in order to monitor the insulation condition of medium- and high-voltage power cables. In this context, an experimental evaluation of three different printed circuit board (PCB)-based inductive sensor topologies, with spiral, non-spiral, and meander shapes, is performed. The aim is to assess their capabilities for PD detection along a transmission power cable. First, simulation and experimental characterization are carried out to determine the equivalent electrical circuit and the quality factor of the three sensors. PD activity was studied in the lab on a 10-m-long defective MVAC cable. The three PCB-based sensors were tested in three different positions: directly on the defective cable (P1), at a separation distance of 10 cm to 3 m (P2), and on the ground line (P3). For the three positions, all sensors’ outputs present a damped sine wave signal with similar frequencies and durations. Experimental results showed that the best sensitivity was given by the non-spiral inductor, with a peak voltage of around 500 mV in P1, 428 mV in P2, and 45 mV in P3, while the meander sensor had the lowest values, which were approximately 80 mV in P1. The frequency spectrum bandwidth of all sensors was between 10 MHz and 45 MHz. The high sensitivity of the non-spiral inductor could be associated with its interesting properties in terms of quality factor and SFR, which are due to its very low resistivity. To benchmark the performance of the designed three-loop sensors, a comparison with a commercial high-frequency current transformer (HFCT) is also made.

## 1. Introduction

The growing global demand for electrical energy, combined with the substitution of fossil fuels, is posing new challenges at many levels of the electric grid, especially concerning the current structure of the transmission network. Large amounts of energy and very high voltages now must be transmitted over increasingly long distances. Due to their advantages, such as high reliability, less interference with neighboring communication lines, safety, and low space occupation, high- and medium-voltage alternating current (HVAC/MVAC) underground transmission cables are replacing overhead lines in many important parts of the electric power system. The structure of the insulated underground transmission cables differs slightly from that of the bare overhead lines, so their preventive maintenance presents a challenging problem [1]. Indeed, extremely small defects may slip through the manufacturer’s quality control procedures and remain within the cable sheath as manufactured, resulting in a partial discharge (PD) phenomenon that subsequently heightens during operation due to environmental conditions and chemical, mechanical, and electro-thermal aging [2,3]. PD is a localized dielectric breakdown caused by an intense electric field exceeding the dielectric strength of the insulation system under high voltage stress. Because these local dielectric breakdowns are limited to a small portion of the insulation (voids, cracks, etc.) and involve low energy, the immediate consequences are not disastrous [4] Nonetheless, in the long term, repetitive physical and/or chemical attacks on the defective solid or liquid insulating medium degrade the overall system by erosion until it breaks down [5]. PDs are crucial in cable systems, as they can eventually lead to electrical treeing and failure, which can affect the reliability of the transmission network (both in terms of cable lengths and accessories) [5,6]. Accordingly, PD monitoring is classified as an essential insulation diagnostic routine that indicates local defects and degradation in high-voltage cables, enabling their detection and location using specific tests carried out using extremely sensitive methods [7,8,9]. PD tests can be implemented online or offline and can detect, locate, and characterize cable insulation faults [2].

Various methods have been developed and used for PD testing, including an electrical method based on the IEC 60270 standard as well as other unconventional techniques. Unconventional techniques rely on physical phenomena associated with PD events such as acoustic pressure waves, light waves, chemical by-products, and electromagnetic (EM) waves [10]. Indeed, an increasingly popular method for PD detection is to detect EM radiation due to electric discharge. The EM radiation consists of a combination of different frequency components up to a few GHz, classified into three bands: the high-frequency band (HF), the very high-frequency band (VHF), and the ultra-high-frequency band (UHF) [11]. EM approaches to detect PD activity have interesting characteristics because they allow noninvasive, continuous, and low-cost monitoring of PD activity; however, some of them have a large size, which makes them bulky when used inside equipment such as power transformers and gas-insulated substations (GIS). Several types of sensors can be used for EM wave detection, such as UHF/VHF antennas [12], VHF capacitive couplers [13], or inductive sensors [14]. Capacitive coupler sensors consist of a pair of electrodes separated by a known distance that detect the electric field created by PDs in shielded power cables. Unlike some electrical methods (such as capacitive couplers or transient earth voltage (TEV) sensors) that require direct contact, the UHF method requires no physical contact with the MV/HV equipment to be monitored [15]. Moreover, among the various PD detection methods, the UHF technique has the advantages of online monitoring and defect classification.

Over the last few decades, different categories of inductive sensors have been developed, including high-frequency current transformer (HFCT) sensors [16], inductive loop sensors (ILSs) [17], and Rogowski coil (RC) sensors [18]. These types of sensors detect the PD pulses flowing along the ground line through the associated magnetic field. They are typically made up of one or more copper turns wrapped around a magnetic or non-magnetic core, which is placed close to (in the case of ILSs) or surrounding (in the case of HFTCs and RCs) the ground line through which the PD pulses propagate. HFCT sensors are often clamped into the grounding conductor to the earth network during online PD measurements [19]. The ILS, also known as the RF loop sensor, consists of a conductive loop placed in parallel to the line that leads the PD pulse stream (in series with the coupling capacitor) [20]. Inductive sensors use Faraday’s law to measure the induced voltage in a loop caused by the magnetic field variations due to partial discharge pulses. Their main advantage is that they do not require galvanic contact with the monitored equipment [21,22]. The measuring frequency range for these types of sensors is 30 kHz to 300 MHz, which covers the HF–VHF band. In [20], the authors proposed a PCB-based ILS formed from a single turn to detect PDs in transformers. The latter exhibits a cutoff frequency response of up to 45 MHz. The ILS sensor was placed in different separation positions, parallel to the ground line through which the PD pulse flows. J. Alfredo et al. [23] compare the ability of the three inductive sensors (HFCT, RC, and ILS) to identify and separate different types of PDs using the chromatic technique. The HFCT sensor had a bandwidth of 1 MHz to 80 MHz and a sensitivity of around 25 dB, while the RC sensor operated in the range of 9 MHz to 60 MHz and had a sensitivity of about 12 dB, and the ILS sensor exhibited drift behavior between 0 and 34.69 MHz with a sensitivity of 17.5 dB. Experimentally, these sensors have been shown to be able to monitor various types of PDs from ground line leakage current.

Loop antennas (or magnetic loops) are a type of radio antenna that do not need to be parallel to the ground connection and can operate in open space at close range to the monitored equipment [24], though ideally, their response is maximum when installed parallel to the EM wave propagation direction. Due to their closed design, the antennas are primarily sensitive to the magnetic field and may function in near- or far-field situations depending on the circumstances [25]. Loop antennas are self-resonant, narrowband sensors. According to Faraday’s law of induction, the induced electromotive force in a receiving loop antenna is precisely proportional to the time rate of change of magnetic flux across the loop area. Their main advantages are a simple design with high efficiency, a compact size, and good characteristics. Rozi and Khayam [26] designed three types of small loop coils: circular, square, and triangular, with a bandwidth of 100 MHz. The largest size of the sensor was 9.55 cm for the circular, 7.5 cm for the square, and 10 cm for the triangular loop. The sensitivity of the circle antenna was the best since the peak-to-peak voltage reached 120 mV at a distance of 5 cm above the PD source.

To exploit the advantages of inductive sensing while being away from the ground line, as in the UHF method, this work aims to evaluate the capabilities of three inductive loops having different topologies for PD sensing and compare their performance with an industrial HFCT sensor, along an MVAC power cable. This paper is organized as follows: in the second section, the three inductive sensor topologies are presented and numerically studied to evaluate their lumped electrical parameters using a finite element method (FEM)-based solver and then characterized using a vector network analyzer. The next part will focus on presenting the experimental setup for PD generation along the MVAC cable. The last section presents the measured results in order to explore the most sensitive topology. The responses of the designed sensors, in two different positions, will be analyzed and compared to those of a common industrial HFCT sensor.

## 2. Inductive Sensors: Design and Characteristics

Three planar inductor topologies, shown in Figure 1, will be fabricated to assess their suitability for PD sensing. To determine the optimal design, the three different square topologies, namely spiral, non-spiral, and meander, were built on a printed circuit board (PCB) and experimentally evaluated. These inductors can be classified into two categories: single-layer and double-layer [27]. The double layer, such as the spiral inductor (Figure 1a), requires two levels of metallization (the spiral and the underpass) and two vias. Single-layer planar inductor topologies, such as meander and non-spiral, require only one metal layer, as shown in Figure 1b,c. The three inductors are made on an FR-4-based board with a relative dielectric constant of 4.4 using a copper thickness of 17 µm. Table 1 lists their geometric dimensions, where *n* is the number of turns, *D* and *d* are the outer and inner diameters, respectively, and *w* and *s* are the track width and spacing, respectively. The three fabricated sensors have the same footprint of 20 × 20 mm^2^.

First, the three inductor topologies are simulated, and their electric characteristics are extracted using Ansys HFSS^®^. Figure 2a shows the exact layout of the spiral inductor placed on an FR-4 substrate and surrounded by an air box. In the simulations, the air box dimensions should be 10 times greater than the structure dimensions, to consider all radiated fields. The boundary conditions are defined at the edge of the box by a radiation limit, which allows the field to propagate infinitely far into space. The two-port lumped-element model of the inductive loop sensor is depicted in Figure 2b. The latter consists of a resistance *R_s_* in series with an inductance *L_s_*, both parallel with a capacitance *C_s_*. The elements *L_s_* and *R_s_* represent the series loop inductance and resistance of the track, respectively, while *C_s_* is the stray capacitance.

Figure 3a illustrates the evolution of the sensor-simulated self-inductance. It can be noted that the designed sensor will behave as an inductor for frequencies below the self-resonance frequency (SFR) and switch to behaving capacitively for frequencies beyond. The SRF is a function of the inter-turn stray capacitance *C_s_* and the self-inductance *L_s_*, and can be calculated as:(1)$$SFR=\frac{1}{2\pi \sqrt{L_{s}\;C_{s}}}$$

Figure 3b shows the simulated quality factor, *Q*, of the spiral, non-spiral, and meander inductive sensors. As shown, the quality factor of the non-spiral increases up to 125 at 170 MHz and then drops to zero at the SFR, which is equal to 1.1 GHz. The spiral shows a maximum quality factor of 75 at 36 MHz and an SFR at 99 MHz, while the meander presents the lowest quality factor of 52 at 200 MHz and has an SFR at 600 MHz.

The inductance parameters were extracted from measurements using a vector network analyzer (Keysight E5061B ENA, 100 kHz to 3 GHz). Table 2 shows the different lumped parameters of all sensors’ topologies, extracted from simulations and measurements. The measured values show quite sufficient agreement with the simulated ones. The difference is due to a ±5% inaccuracy in the resulting tracks’ width during fabrication. As we can see, the spiral inductor has the highest inductance value, due to the strong magnetic field flux produced between its spiral tracks. However, the latter clearly exhibits a high value of parasitic parameters (*R_s_* and *C_s_*), which have a significant impact on the resonant frequency compared to the non-spiral one. Despite its very low inductance, the non-spiral inductor shows interesting properties in terms of quality factor and SFR thanks to its very low resistivity. Lastly, the meander sensor exhibits a higher inductance value compared to the non-spiral sensor and a higher SFR compared to the spiral layout thanks to the low inter-turn stray capacitance despite the high resistive losses.

## 3. Defective Cable and Experiment Setup

### 3.1. Power Cable Characteristics

To evaluate the performance of the designed PCB-based magnetic sensors, a 10-m-long MVAC single-core power cable (rated voltage: 10/12 kV) with a conductor cross-section of 150 mm^2^, manufactured according to the IEC-60502-1 standard, was used as the PD source. Figure 4a shows the stacking of the layers (insulators and conductors) forming the MV cable, and Figure 4b shows the location of the defect in a cross-sectional view. Table 3 lists the cable characteristics and composition. The insulation layer was accidentally manufactured with internal defects located 10 cm from one end of the cable marked on the outside by protrusions and delamination of the sheath.

### 3.2. PD Assessment Setup

The laboratory generation setup of the PDs is shown in Figure 5, and its equivalent electrical connection circuit is illustrated in Figure 6. It consists mainly of a 100 kV transformer, a capacitive voltage divider, and a defective power cable. A DC voltage regulator with a range of 0–400 V was utilized to set the required AC–HV applied voltage. A 50 Hz AC source from the test transformer plays a crucial role in supplying the nominal voltage in the system. The output voltage of the transformer would be raised through the DC regulator until the PD occurred, which would be marked by clearly distinguishable PD patterns on the PC monitor. As shown in Figure 5, the two terminals of the defective cable were connected to the HVAC generator. The AC voltage is applied to the core conductor, while the screen of the cable is grounded. In this test, PCB-based sensors were assessed in three different positions: Position 1, where the sensors were placed directly on the defect (P1 in Figure 6), then at a variable distance from 10 cm up to 3 m away from the defect (P2 in Figure 6), and lastly Position 3, where the sensors were placed on the wire ground at a position similar to the HFCT (P3 in Figure 6). In this experiment, the three PCB-based inductive topologies, i.e., spiral (Figure 7a), non-spiral (Figure 7b), and meander (Figure 7c) were experienced at these three positions. The commercial HFCT (140–100 HVPD) with a diameter of 140 mm, shown in Figure 7d, was used to benchmark the recorded performance of the designed PCB-based loop antennas. All voltage outputs were recorded with a PicoScope 3000 Series (sampling frequency: 1 GS/s, 8-bit, and 100 MHz bandwidth) when using a 50 Ω coaxial cable. The PicoScope was connected to the computer using a USB cable to acquire and store the data from the PD measurement.

## 4. PD Measurement: Results and Discussion

At first, a PD measurement under AC voltage was carried out to characterize the type of discharge associated with the faulty cable. This experiment was performed with a gradually increasing voltage to determine the value of the PD Inception Voltage (PDIV) under AC stress. The HFCT sensor is clamped around the ground cable. During the test, it turned out that the magnitude of the applied voltage that caused PDs was 8.2 kV, which corresponds to the PDIV. To identify the type of PD and the noise level acquired during the test, the phase-resolved partial discharge (PRPD) of the HFCT sensor, reported in Figure 8, should be analyzed. Hence, more than 3000 pulses of PDs were acquired to obtain statistically significant results. The red curve in the pattern is the reference of the AC-applied voltage and is useful for defining the phase of the acquired PDs. The reported PRPD patterns have been extracted from the PryCam Grids acquisition software available in the lab. As can be seen from its shape, the specimen exhibits the presence of internal discharges. It is worth noting that all the measurement sensors used in this experiment (i.e., spiral, non-spiral, and meander) exhibit a PRPD similar to that of the HFCT. It is well known that internal PD is the most common leading cause of premature insulator failure in power cables. This proves the defective aspect of the used cable, since air microcavities are trapped in the dielectric material and, therefore, when the insulator is subjected to an external electric field, a difference is created between the electric field of the air present in the microcavity and that of the surrounding dielectric layer.

### 4.1. Induced Voltages Shapes and Spectra

Next, the induced voltage generated by the three designed inductive sensors will be recorded. As expounded, during the test, the three sensors will be placed in three different positions:

Position 1 (P1 in Figure 6): the PCB-based inductors were placed directly on the defect. Figure 9, Figure 10 and Figure 11 present the induced voltage detected by the spiral, non-spiral, and meander sensors and their frequency spectra. As shown, all sensors’ outputs present a damped sine wave signal with different frequencies and durations. Figure 9a,b depict, respectively, the time-domain PD signal waveforms and their frequency spectrums detected by the inductive spiral sensor. As seen, the sinusoidally damped signal has a peak voltage of ∼100 mV, which was recorded by the PicoScope without any amplification circuit. The corresponding frequency spectrum of the induced voltage is drawn in Figure 9b. The peak spectrum frequency was around ∼32 MHz. The rise time of the PD pulses was typically about 0.22 µs.

Figure 10a,b show, respectively, the induced voltage waveform detected by the non-spiral inductor and its frequency spectrum. As seen here, the maximum amplitude was about 500 mV with a rise time of 0.22 µS. It is worth noting that the resulting pulse detected by the non-spiral sensor presents an offset of ~75 mV, and this is most likely due to the presence of low-frequency (50 Hz) content noise. The spectrum frequency shows a main peak at ~32 MHz. Lastly, Figure 11a,b show, respectively, the induced voltage signal detected by the meander sensor and its spectrum frequency. The maximum amplitude of the detected PD pulse was approximately 80 mV. The frequency spectrum of the detected signal, shown in Figure 11b, displays the fundamental peak at ~32 MHz. It could be clearly noted that the meander sensor exhibits the lowest sensitivity for PD detection and the lowest number of oscillations (lower PD pulse duration), while the non-spiral inductor presents the highest sensitivity with almost a single-frequency spectrum.

Position 2 (P2 in Figure 6): Next, all sensors were moved 10 cm away from the defect. Figure 12a,b show, respectively, the PD signal pulse detected by the spiral sensor and its frequency spectrum. As seen, the sensitivity of the sensor decreases and shows a maximum amplitude of 79 mV. Figure 13a,b illustrate, respectively, the frequency spectrum of the PD signal detected by the non-spiral and meander sensors. The non-spiral inductor presents a maximum pulse amplitude of 428 mV, and the meander sensor shows a maximum amplitude of 57 mV. From the results of the three sensors, it is obvious that when placing the sensor 10 cm away from the defect, its sensitivity decreases. The non-spiral sensor presents the highest sensitivity in this position as well, while the meander sensor shows the lowest performance.

To investigate the attenuation of PD pulse detection over distance, the non-spiral sensor was placed at variable distances from 10 cm up to 3 m away from the defective cable. Table 4 shows the amplitude and the fundamental peak frequency of the PD pulse versus distance. As expected, the amplitude of the induced voltage decreases as the distance from the defect increases. The main peak frequency of the pulse remains almost the same.

Position 3 (P3 in Figure 6): In this position, the three sensors were placed on the ground wire. Figure 14 and Figure 15 show the induced voltage of the PD pulse and its frequency spectrum detected by the spiral, non-spiral, and meander sensors. Results show that the spiral sensor detects a maximum amplitude of about 30 mV (Figure 14a) and presents a main spectrum frequency peak at 18 MHz (Figure 14b), while the non-spiral sensor presents a maximum pulse amplitude of 45 mV. The non-spiral sensor shows a main frequency band starting at 25 MHz and going up to 45 MHz (Figure 15a), which corresponds to the frequency band of the PD signal detected by the non-spiral sensor. The frequency components beyond 32 MHz are comparatively small. Lastly, the maximum amplitude of the PD pulse detected by the meander sensor was about 18 mV, and the frequency spectrum of the signal detected shows the main frequency components at 25 MHz and 37 MHz (Figure 15b). Again, as can be noted, the non-spiral sensor presents a higher sensitivity of PD detection than the spiral inductive sensor in both positions, and it presents a smaller number of oscillations in the signal than the spiral sensor. It is clearly noted that the meander sensor presents the lowest sensitivity for PD detection compared to the spiral and non-spiral sensors in the two positions.

To evaluate the sensors’ sensitivity, we compared the responses of the three designed sensors with a commercial HFCT sensor, “HFCT Ø140”. The choice of the HFCT sensor results from the fact that it is also based on inductive detection. Figure 16a shows that the latter provides a voltage amplitude of around 250 mV with few oscillations after the first peak. The spectral frequency of the signal shows a high-frequency component at about ~1 MHz (Figure 16).

### 4.2. Notes and Observations

Through DPs monitoring, the dielectric condition of high-voltage equipment can be assessed, and insulation defects can be detected and located. HFCT sensors detect the presence of partial discharges, arcs, or sparks by detecting high-frequency pulses flowing in the ground wire [28]. Their basic principle is to produce an induced voltage proportional to the fast transient current created by the PD occurrence. Nevertheless, HFCT sensors cannot accurately locate the damaged part of an insulated system. On the other hand, the UHF method could be used for online PD monitoring and can help localize the defects [23]. Indeed, it is through the relationship between the arrival time and/or amplitude and/or frequency of the output wave from the UHF antenna and the separation distance that the PD can be located (typically three sensors are needed). Since EM waves in space consist of an electric field measured in V/m and a magnetic field measured in A/m, by using an inductive loop, the magnetic field perpendicular to the electric field can be detected [24]. Inductive loops are based on Faraday’s law of induction to measure changes in the magnetic field caused by the PD propagated in the air surrounding the defect. The variation of the magnetic field induces an electric vector field, which gives rise to a potential difference or induced voltage, *e*, defined as:(2)$$e=\oint \vec{E}.\vec{dl}=-nS\frac{dB}{dt}$$
where *E* is the electric field, *n* is the number of turns of the inductor, *B* is the magnetic field, and *S* is the loop inductor’s area.

The major advantage of the designed inductive loop is that it allows a measurement of the swiftly varying currents caused by the PD when placed in position P3 (as the HFCT sensor), as well as the magnetic field propagating in close space (up to 3 m) as UHF sensors. In fact, since the PD phenomenon produces a very short current pulse that travels through the main ground conductor, this leads to a change in the corresponding magnetic field, which connects the secondary of each sensor, coil, or loop and induces a voltage *e* proportional to the rate of change of the current *i*. The transfer function of these sensors can be expressed as follows:(3)$$e=M\frac{di}{dt}$$
where *M* is the proportional constant or the mutual link between the earth conductor and the inductive loop (the secondary). Again, the behavior of the sensors is based on Faraday’s law. 

In conclusion, the three inductive sensors printed on a PCB with different topologies showed sufficient capability for PD detection on the tested power cable. As expected, HFCT are bulky sensors with ferromagnetic cores, so they have higher sensitivity than the designed inductors, and, as a consequence, the output signal, with a common bandwidth of 500 kHz–50 MHz, is proportional to the current at low frequencies [16]. The non-spiral sensor outperforms the spiral and meander sensors thanks to its high-quality factor. Inductive sensors have several advantages over HFCT sensors, including their small size, simplicity, low cost, and lack of the need for an external power supply. In addition to clamping to the ground wire, they can be placed in the open air near the PD source to allow its location. As a result, the inductive loop could be useful for measuring EM waves generated by PD activities as well as pulses flowing in the ground. The detection of PD with inductive loops could be used in many types of equipment, such as gas-insulated substations (GIS), air-insulated substations (AIS), power transformers, insulators in aerial distribution cables, joints and splices of insulated cables, and rotating machinery.

## 5. Conclusions

In this work, the performance toward PD detection of three PCB-based inductive sensor topologies, named spiral, non-spiral, and meander, was investigated. Initially, the electrical parameters of the three topologies were examined using FEM simulation and experimental measurement. The non-spiral sensor outperformed the spiral sensor in terms of quality factor, with a value of 125 and a self-resonance frequency (SFR) of 1.1 GHz. The three sensors were experimentally tested for PD detection on a 10-m-long power cable in three different positions: (P1) directly on the defective cable, (P2) at a separation distance of 10 cm to 3 m, and (P3) on the ground line. The PDIV measurements of the three sensors are relatively the same as the HFCT sensor, with a value of 8.2 kV. The experiment results show that for the three positions, the non-spiral inductive sensor has the highest sensitivity for PD detection, while the meander sensor has the lowest. The frequency spectrum bandwidth of the three sensors’ outputs is located between 10 MHz and 45 MHz, which is quite similar to the HFCT sensor. In conclusion, it could be confirmed that the proposed planar inductive loops exhibit several advantages over HFCT sensors, such as small size, simplicity, low cost, and remote operation, and seem to be a suitable and promising candidate for PD monitoring in the high-voltage power cable, where irradiated magnetic fields should potentially be monitored.

## Figures and Tables

**Figure 1 sensors-23-00290-f001:**
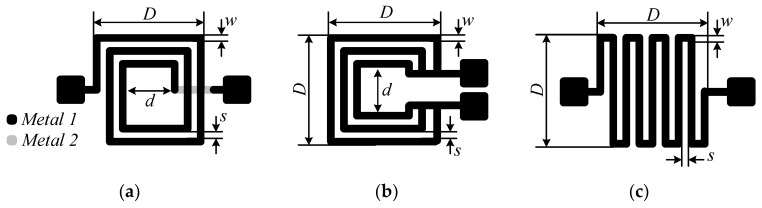
Investigated planar inductive sensor topologies: (**a**) spiral, (**b**) non-spiral, and (**c**) meander.

**Figure 2 sensors-23-00290-f002:**
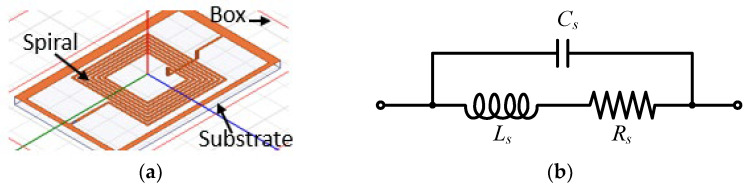
The spiral inductor topology: (**a**) designed in HFSS, and (**b**) the two-port lumped-element model.

**Figure 3 sensors-23-00290-f003:**
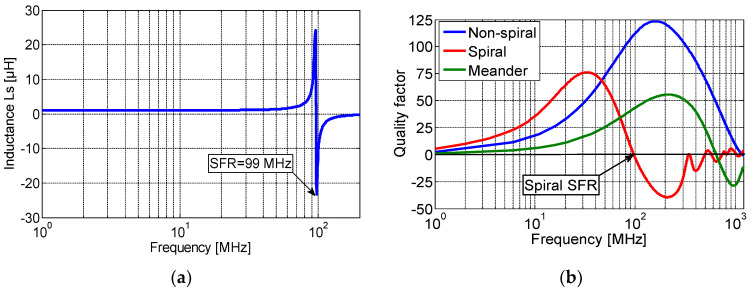
(**a**) The simulated inductance variation of the spiral sensor and (**b**) the simulated quality factor of the three designed inductive sensor topologies.

**Figure 4 sensors-23-00290-f004:**
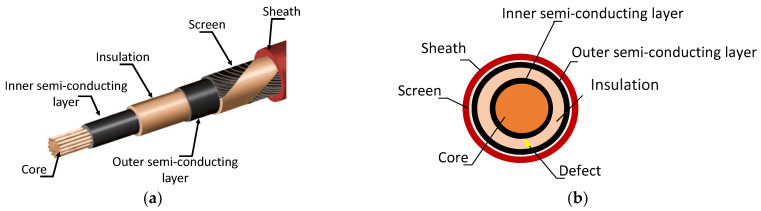
Defective used cable to generate PD: (**a**) layers stacking and (**b**) cross-section (indicating defect location).

**Figure 5 sensors-23-00290-f005:**
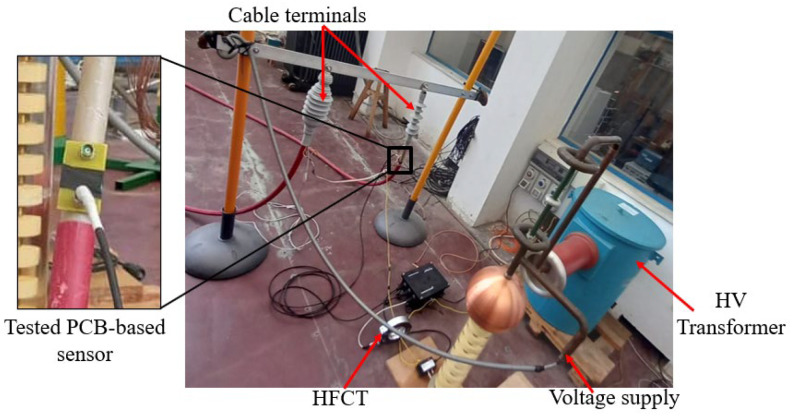
Measurement laboratory setup of PDs in the MV power cable (enlarged image is showing the spiral inductor in position P1).

**Figure 6 sensors-23-00290-f006:**
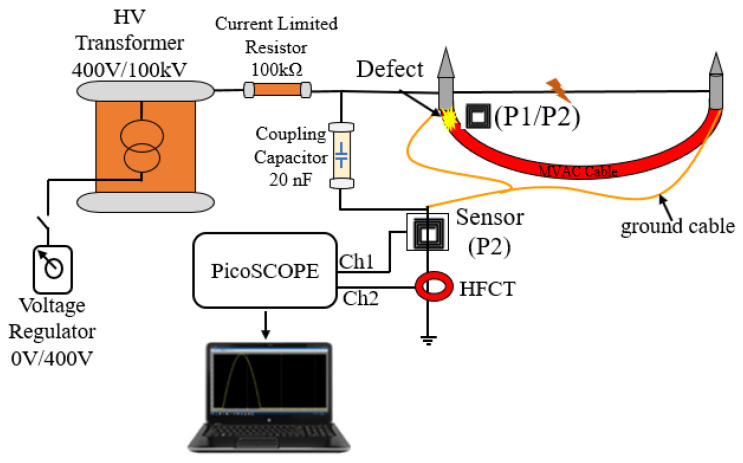
Equivalent electrical connection circuit of the setup.

**Figure 7 sensors-23-00290-f007:**
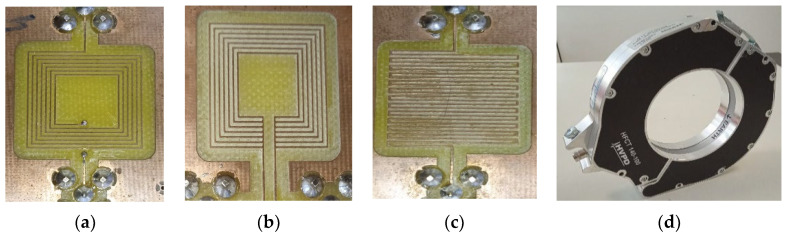
Inductive sensors tested in the experiment setup: (**a**) spiral, (**b**) non-spiral, (**c**) meander, and (**d**) the HFCT.

**Figure 8 sensors-23-00290-f008:**
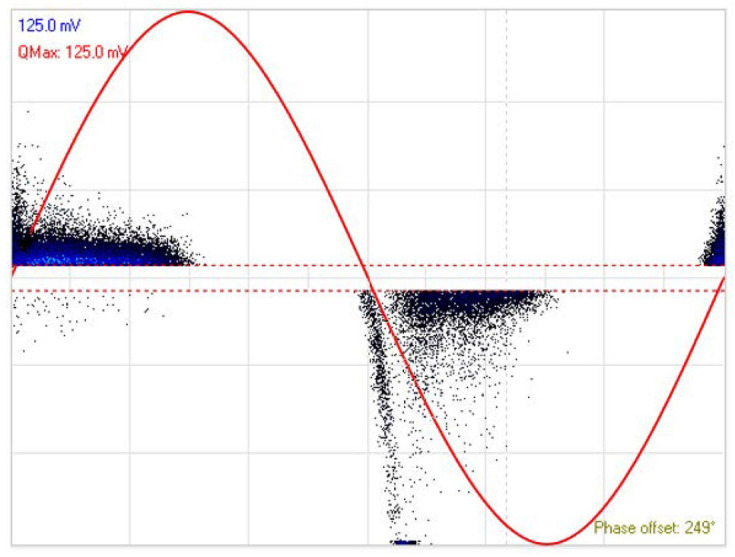
PD patterns detected by the HFCT sensor, under an AC-applied voltage of 8.2 kV (red).

**Figure 9 sensors-23-00290-f009:**
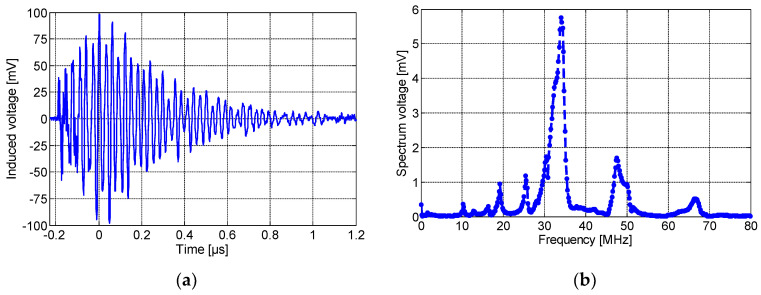
(**a**) The time-domain wave due to partial discharges at 8.2 kV, and (**b**) its frequency spectrum, detected by the inductive spiral sensor placed at P1.

**Figure 10 sensors-23-00290-f010:**
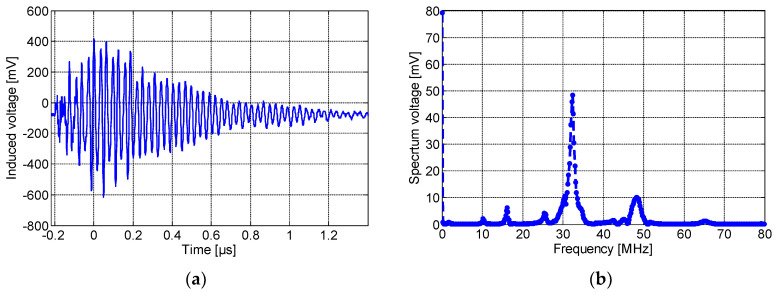
(**a**) The time-domain wave due to partial discharge at 8.2 kV and (**b**) its frequency spectrum, detected by the non-spiral inductive sensor placed on P1.

**Figure 11 sensors-23-00290-f011:**
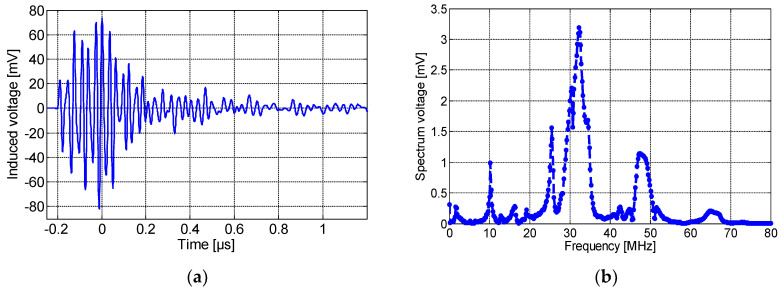
(**a**) The time-domain wave due to the partial discharge at 8.2 kV and (**b**) its frequency spectrum, detected by the meander sensor placed on P1.

**Figure 12 sensors-23-00290-f012:**
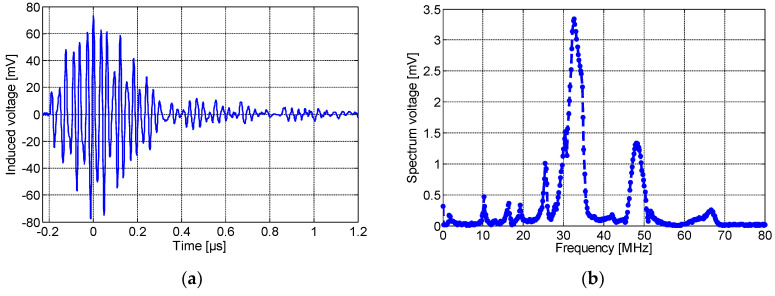
(**a**) The time-domain wave due to partial discharges at 8.2 kV and (**b**) its frequency spectrum detected by the spiral when placed at 10 cm away from the defective cable.

**Figure 13 sensors-23-00290-f013:**
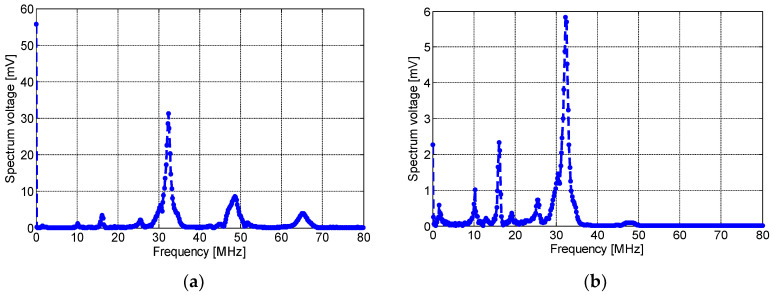
Frequency spectra detected by the: (**a**) non-spiral, and (**b**) meander, inductive sensors when placed at 10 cm away from the defective cable.

**Figure 14 sensors-23-00290-f014:**
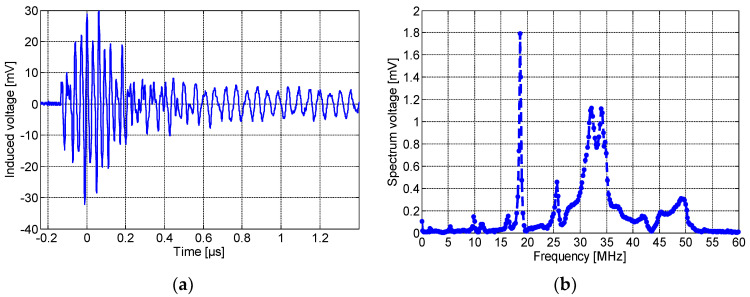
(**a**) The time-domain wave of the output voltage at 8.2 kV and (**b**) its frequency spectrum detected by the spiral sensor placed at P3.

**Figure 15 sensors-23-00290-f015:**
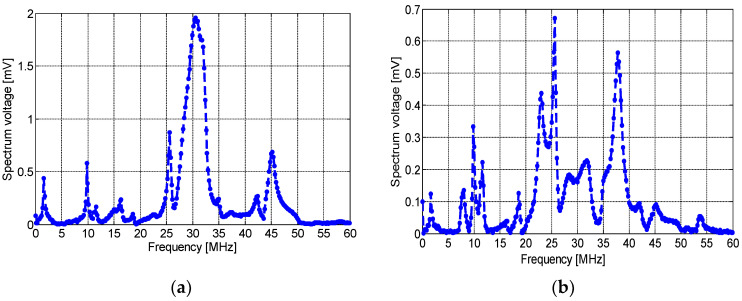
Frequency spectra detected by the: (**a**) non-spiral and (**b**) meander, inductive sensors when placed at P3.

**Figure 16 sensors-23-00290-f016:**
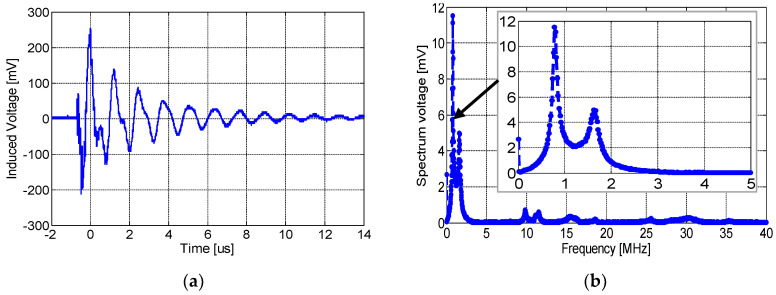
(**a**) The time-domain wave of the output voltage at 8.2 kV and (**b**) frequency spectrum of the HFCT sensor.

**Table 1 sensors-23-00290-t001:** Geometrical dimensions of the PCB-based manufactured inductors.

Parameter	*n*	*D*	*d*	*w*	*s*
Value	6.5	20 mm	5.2 mm	0.4 mm	0.4 mm

**Table 2 sensors-23-00290-t002:** Simulated/measured lumped parameter values of the different sensor topologies.

Parameters/ Topology	Ls (nH) (@1 MHz)		Rs (Ω) (@1 MHz)		Cs (pF)		SFR (MHz)	
	Simul	Meas	Simul	Meas	Simul	Meas	Simul	Meas
Spiral	997	1000	1.03	0.95	1.3	1.6	99	130
Non-spiral	19	22	0.04	0.06	1.1	1.06	1100	1085
Meander	190	185	0.99	1.3	0.59	0.62	608	463

**Table 3 sensors-23-00290-t003:** Cable composition and characteristics.

Cable Parameters	Characteristics
Core	Conductor Material: Copper
Conductor cross-section	150 mm^2^
Inner semi-conducting layer	Black semi-conductive cross-linkable compound
Insulation	Hard-grade ethylene propylene rubber (HEPR)
Insulation thickness	4.85 mm (min.)/5.5 mm (avg.)
Outer semi-conducting layer	Extruded cold strippable elastomeric compound
Earth screen	Bare copper wire
Earth screen cross-section	25 mm^2^
Outer sheath	Special thermoplastic, M16 type (red color)

**Table 4 sensors-23-00290-t004:** PD pulse amplitude and peak frequency detected by the non-spiral sensor as a function of distance from the defect.

Separation distance (cm)	0 (P1)	10 (P2)	20	30	40	50	100	150	300
PD pulse amplitude (mV)	500	428	375	334	296	258	216	208	71
Main peak Frequency (MHz)	34	32	32.6	32.4	32.3	32.3	32.3	32.2	15.9

## Data Availability

The data presented in this study are available on request from the corresponding author. Some results are discussed but not drawn due to page limitations.

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
