# Peer review of "PCB-Based Planar Inductive Loops for Partial Discharges Detection in Power Cables"

_sensors, 2022, doi:10.3390/s23010290_

Round 1
Reviewer 1 Report
Comments:
1. Introduction chapter is correct written and planned. I expected to find some information in this chapter about actual state of research on the world. I have read some papers, where that kind of investigations was presented. So, please complete this chapter, and say what authors improved on the basis of their study.
2. Used references are correct. I did not notice auto citations practice.
3. Chapter 2 – there is some mistake of references – Errors! I did not find references in the chapter. Probably it is some format mistake.
4. Chapter 2 – Authors investigated various shape of sensor electrode to detect PD. I would expect some fundamental, physical information, principles, which induced authors to choose studied geometry of mentioned sensors. What was a reason to choose exactly the shape of sensors, which were studied. Please complete. I think, some fundamentals of antenna theory is needed.
5. Chapter 2 – comparison to state of research is needed. It is difficult to say what authors added to general state of research.
6. Fig.4 – is it figure made by authors? If not, reference is lost.
7. Fig.4 – what cable was it? What voltage was it? 110, 220, 400 kV?
8. Chapter 3 – what exactly defect was simulated in the lab? Please describe using more details. Please complete.
9. Chapter 3 – what criteria were used to assess the various propositions of mentioned sensors it is very important. Please complete.
10. General opinion. The topic is very interesting and important. I think, the paper should be completed, because many details are lost.
Author Response
First of all, we appreciate the helpful comments and suggestions given by the reviewer on our manuscript. The latter has been revised accordingly, with changes in the original text marked with Yellow Fluorescent. Below are the responses provided to the respectable reviewer that, we hope, will meet his expectations.
******************************
Point 1: Introduction chapter is correct written and planned. I expected to find some information in this chapter about actual state of research on the world. I have read some papers, where that kind of investigations was presented. So, please complete this chapter, and say what authors improved on the basis of their study
Response: We totally agree with the reviewer's point of view, it is for this reason that the whole introduction has been redone, and it is marked with the yellow fluorescent in the new version. In summary, the main contribution of this design is that we proved, for the first time, that Inductive loops are able to detect the presence of partial discharges in the power cables. In addition, the major advantage of the designed inductive loop is that it allows a measurement of the swiftly varying currents caused by the PD when placed in position P3 (as the HFCT sensor), as well as the magnetic field propagating in close space (up to 3 m) as UHF sensors.
Point 2: Used references are correct. I did not notice auto citations practice.
Response: Thank you for your respectful comment.
Point 3: Chapter 2 – There is some mistake of references – Errors! I did not find references in the chapter. Probably it is some format mistake.
Response: Indeed, when the editorial board transferred the manuscript to the Sensors template, all the hypertext links (mainly images) were lost. We are sorry, we fixed this issue in this new version.
Point 4: Chapter 2 – Authors investigated various shape of sensor electrode to detect PD. I would expect some fundamental, physical information, principles, which induced authors to choose studied geometry of mentioned sensors. What was a reason to choose exactly the shape of sensors, which were studied? Please complete. I think some fundamentals of antenna theory are needed.
Response: The chosen topologies are the most common in the bibliography for planar inductance. Indeed, each one has a different characteristic from the others. The spiral has a large self-inductance because of its spiral shape (we will get an average quality factor Q), the meander has a low inter-turn stray capacitance (low Q), and the non-spiral has a very low resistance because of its loops in parallel (a large quality factor Q). These three forms provide a good variety for comparing results.
Regarding antenna theory, we further developed the corresponding paragraph in Section 4.2 to explain the operation of the inductive loops in the three positions tested: P1, P2, and P3. The principle is always based on Faraday's law of induction, except that in positions P1 and P2, the variable magnetic field comes directly from the PD, while in position P3, it comes from the variable current flowing through the ground wire.
Point 5: Chapter 2 – comparison to state of research is needed. It is difficult to say what authors added to general state of research.
Response 5: We developed the introduction further to emphasize our contribution to the state of the art. In summary, a new partial discharge sensor has been introduced that has the advantage, compared to HFCT sensors, of being smaller, simpler, cheaper, and not requiring an external power supply. In addition to clamping to the earth wire, it can be placed in the open air near the PD source to allow it to be located. So it could combine the advantages of HFCT sensors and UHF antennas.
Point 6: Fig.4 – is it figure made by authors? If not, reference is lost.
Response 6: Figure 4 is drawn by the authors based on the cable datasheet.
Point 7: Fig.4 – what cable was it? What voltage was it? 110, 220, 400 kV?
Response 7: We added this sentence in the text for more details:
“To evaluate the performance of the designed PCB-based magnetic sensors, a 10-meter-long MVAC single-core power cable (rated voltage: 10/12 kV), with a conductor cross-section of 150 mm2, manufactured according to the IEC-60502-1 standard, was used as the PD source.”
Point 8: Chapter 3 – what exactly defect was simulated in the lab? Please describe using more details. Please complete.
Response 8: It is an internal fault in the cable; we do not know exactly its nature since this kind of fault should not occur intentionally. It is most likely an air bubble in the insulation that slipped during manufacturing. We know in advance that the cable is faulty at this exact location and we used it for real-world testing.
Point 9: Chapter 3 – what criteria were used to assess the various propositions of mentioned sensors it is very important. Please complete.
Response 9: The main criterion used to compare the different topologies is the peak voltage. The better it is, the better the sensor is, and will be able to detect partial discharges of lower intensity and also will not need complicated processing electronics in a later stage.
Thank you so much for your helpful comments.
Reviewer 2 Report
sensors-2101944
PCB-based Planar Inductive Loop for Partial Discharges Detection in Power Cables
The manuscript is well-written and easy to follow.
Some points need to be known.
-Reviewers cannot refer to any figure as there is an error "Error! Reference source not found" against all figures.
- Can we use flexible PCB in place of FR4? Maybe it will give much better results?
- Are there may need some other configurations except (a) spiral, (b) non-spiral, and (c) meander?
- In Table 2, please remove @ symbol.
- It will be good to compare the proposed method with other existing methods.
Author Response
First of all, we appreciate the helpful comments and suggestions given by the reviewer on our manuscript. The latter has been revised accordingly, with changes in the original text marked with Yellow Fluorescent. Below are the responses provided to the respectable reviewer that, we hope, will meet his expectations.
******************************
Point 1: Reviewers cannot refer to any figure as there is an error "Error! Reference source not found" against all figures.
Response 1: Indeed, when the editorial board transferred the manuscript to the Sensors template, all the hypertext links (mainly images) were lost. We are sorry, we fixed this issue in this new version.
Point 2: Can we use flexible PCB in place of FR4? Maybe it will give much better results?
Response 2: Yes, we believe that a flexible PCB sensor may perform better for the P3 position if the inductor wraps perfectly around the ground wire. This will be the subject of forthcoming papers on the same subject. However, the use of flexible PCB will not change anything for positions P1 and P2.
Point 3: Are there may need some other configurations except (a) spiral, (b) non-spiral, and (c) meander?
Response 3: The chosen topologies are the most common in the bibliography for planar inductance. Indeed, each one has a different characteristic from the others. The spiral has a large self-inductance because of its spiral shape (we will get an average quality factor Q), the meander has a low inter-turn stray capacitance (low Q), and the non-spiral has a very low resistance because of its loops in parallel (a large quality factor Q). These three forms provide a good variety for comparing results.
Point 4: In Table 2, please remove @ symbol.
Response 4: The sign @ refers to ‘at’, and is mandatory to specify the frequency at which the value is measured. Hence, all the lumped parameters are measured at the frequency of 1 MHz.
Point 5: It will be good to compare the proposed method with other existing methods.
Response 5: We totally agree with the reviewer's point of view, it is for this reason that the whole introduction has been redone, and it is marked with the yellow fluorescent in the new version. In summary, the main contribution of this design is that we proved, for the first time, that Inductive loops are able to detect the presence of partial discharges in the power cables. In addition, the major advantage of the designed inductive loop is that it allows a measurement of the swiftly varying currents caused by the PD when placed in position P3 (as the HFCT sensor), as well as the magnetic field propagating in close space (up to 3 m) as UHF sensors.
Thank you so much for your helpful comments.
Round 2
Reviewer 1 Report
The authors have complied with all my comments. The article contains all my suggestions. The authors answered all my questions correctly. I think the article can be published in its current form.